# Rapid assessment of avoidable blindness and cataract surgery coverage among forcibly displaced Myanmar Nationals (Rohingya refugees) in Cox's Bazar, Bangladesh

A. H. M. Enayet Hussain[1], Munir Ahmed[2], Jerry E. Vincent[3,4,5]*, Johurul Islam[6], Yuddha D. Sapkota[7], Taraprasad Das[8,9], Nathan Congdon[10], Lutful Husain[2], Gulam Khandaker[6,11,12,13,14], Mohammad Muhit[6,11], Mohammad Awlad Hossain[10], Danny Haddad[10]

1 Directorate General of Medical Education, Ministry of Health and Family Welfare, Dhaka, Bangladesh, 2 Orbis Bangladesh, Dhaka, Bangladesh, 3 Seva Foundation, Berkeley, California, United States of America, 4 Faculty of Optometry, Rangsit University, Pathum Thani, Thailand, 5 Faculty of Public Health, Thammasat University, Pathum Thani, Thailand, 6 CSF Global, Dhaka, Bangladesh, 7 International Agency for the Prevention of Blindness Southeast Asia, Kathmandu, Nepal, 8 International Agency for the Prevention of Blindness Southeast Asia, Hyderabad, India, 9 L V Prasad Eye Institute, Hyderabad, India, 10 Orbis International, New York, New York, United States of America, 11 Asian Institute of Disability and Development (AIDD), University of South Asia, Dhaka, Bangladesh, 12 School of Health, Medical and Applied Sciences, Central Queensland University, Rockhampton, Queensland, Australia, 13 Discipline of Child and Adolescent Health, Sydney Medical School, The University of Sydney, Sydney, Australia, 14 Central Queensland Public Health Unit, Central Queensland Hospital and Health Service, Rockhampton, Queensland, Australia

* jerry.vincent@outlook.com

**Data Availability Statement:** The dataset for this study is archived in the RAAB repository (http://raabdata.info/).

## Abstract

### Aim

To determine the prevalence and causes of blindness, vision impairment and cataract surgery coverage among Rohingya refugees aged ≥ 50 years residing in camps in Cox's Bazar, Bangladesh.

### Methods

We used the Rapid Assessment of Avoidable Blindness (RAAB) methodology to select 76 clusters of 50 participants aged ≥ 50 years with probability proportionate to size. Demographic and cataract surgery data were collected using questionnaires, visual acuity was assessed per World Health Organization criteria and examinations were conducted by torch, and with direct ophthalmoscopy in eyes with pinhole-corrected vision <6/12. RAAB software was used for data entry and analysis.

### Results

We examined 3,629 of 3800 selected persons (95.5%). Age and sex adjusted prevalence of blindness (<3/60), severe visual impairment (SVI; >3/60 to ≤6/60), moderate visual impairment (MVI; >6/60 to ≤6/18), and early visual impairment (EVI; >6/18 to ≤6/12) were

**Funding:** This study was funded by the Non-Communicable Disease Centre (NCDC) of the Directorate General of Health Services (DGHS). The funder had no role in study design, data collection and analysis, decision to publish or preparation of the manuscript.

**Competing interests:** The authors have declared that no competing interest exist.

2.14%, 2.35%, 9.68% and 14.7% respectively. Cataract was responsible for 75.0% of blindness and 75.8% of SVI, while refractive error caused 47.9% and 90.9% of MVI and EVI respectively. Most vision loss (95.9%) was avoidable. Cataract surgical coverage among the blind was 81.2%. Refractive error was detected in 17.1% (n = 622) of participants and 95.2% (n = 592) of these did not have spectacles. In the full Rohingya cohort of 76,692, approximately 10,000 surgeries are needed to correct all eyes impaired (<6/18) by cataract, 12,000 need distance glasses and 73,000 require presbyopic correction.

## Conclusion

The prevalence of blindness was lower than expected for a displaced population, in part due to few Rohingya being ≥60 years and the camp's good access to cataract surgery. We suggest the United Nations High Commissioner for Refugees include eye care among recommended health services for all refugees with long-term displacement.

## Introduction

The World Health Organization's (WHO) World Report on Vision states that 2.2 billion people are affected by blindness or vision impairment (BVI) and that this number is likely to increase over the coming decades due principally to population growth and aging [1]. International efforts to address BVI call for reducing rates of blindness by 25% and assuring that universal eye care is accessible and affordable for all people everywhere, using an integrated people-centered approach [1, 2]. Low and middle-income countries, women, rural-dwellers, migrants, indigenous peoples, and persons with disabilities are more likely to be affected by BVI [1, 3–5]. Studies also suggest that crisis-affected populations such as refugees are likely to be disproportionately affected, however, findings from the limited number of available studies are inconsistent [6–10].

The Rohingya are a minority ethnic group residing in the Rakhine state of Myanmar. According to UN estimates, in late September 2017, approximately 10,000 Rohingya were killed and over 730,000 fled into Bangladesh [11]. These refugees joined over 200,000 Rohingya displaced by previous violence [12]. The United Nations High Commissioner for Refugees (UNHCR) now reports over 919,000 Rohingya refugees are residing in several camps in the two southernmost sub-districts of Cox's Bazar district [13]. One of these camps, Kutupalong (population over 600,000), is now the largest refugee camp in the world [14]. The Bangladesh government refers to Rohingya refugees as Forcibly Displaced Myanmar Nationals (FDMNs).

Cox's Bazar District is a remote area with limited availability of eye care services. Cox's Bazar Baitush Sharaf Hospital (CBBSH), a social enterprise eye care facility, is the only ocular surgical facility in the district [15] and serves the 2.3 million local residents in addition to the refugees [16].

At the onset of this crisis, several non-governmental organizations (NGOs), members of the International Agency for the Prevention of Blindness (IAPB) working in Bangladesh, agreed to collaborate in a response to the crisis. Under a memorandum of understanding with the Ministry of Health and Family Welfare, these agencies created the Cox's Bazar Eye Response Working Group to strengthen the eye health system in Cox's Bazar district and better serve both the local population and refugees [17]. Current data on the burden and causes of

BVI are needed for evidence-based planning and development of eye services, but such data are currently lacking for the Rohingya refugees.

In this study we aimed to estimate the prevalence and causes of BVI and to assess the coverage of cataract surgical services among Rohingya refugees.

## Methodology

The Rapid Assessment of Avoidable Blindness (RAAB) is a standardized cross-sectional survey methodology designed to measure the rates and common causes of BVI in participants aged ≥50 years [18]. The RAAB approach has been used in over 300 surveys to date and is faster and less resource-intensive than traditional blindness studies that cover all ages and assess all causes [19]. In this study we used the RAAB survey version 6.0 protocols and software (available at https://www.cehjournal.org/resources/raab/) [20].

### Study population

This study was conducted among the 911,000 Rohingya refugees residing in over 30 UN camps in Cox's Bazar district, Bangladesh. There were 76,962 camp residents aged ≥50 years according to the UN, all of whom were eligible for participation in the study.

### Sample size

In the absence of available data, we assumed a 4.0% prevalence of blindness, anticipating that the Rohingya would have a higher burden than the 3.1% reported in a recent RAAB survey in the Cox' Bazar host population [21]. Using a 20% tolerable error, 95% confidence interval, design effect of 1.5 and an estimated 10% non-response rate, we calculated a sample size of 3,780 (rounded up to 3,800), or 76 clusters of 50 subjects aged ≥50 years.

### Sample selection

The RAAB uses a two-stage cluster-sampling methodology. All refugee camps were included in the sampling frame. Clusters were selected with probability proportionate to size, while households were chosen using systematic compact segment sampling based upon Mahji (refugee leader) administrative areas. Each Mahji's area typically includes about 100 households of approximately 500 people, generally yielding one study cluster of 50 adults aged ≥50 years.

### Public involvement

Local leaders were informed in advance of the study and the purpose and protocols were discussed with local government, UN and refugee officials. Residents of each study cluster were reminded by the Mahji the day before their cluster was surveyed to stay at home and were encouraged to participate. The Mahji for each cluster joined the survey team, helped identify the segment boundaries and assisted with survey processes. Survey results were shared with local authorities, health system agencies, refugee leadership, and the larger refugee community.

### Data collection

Survey data was collected in April and May 2019 by two survey teams. The survey teams visited each study household to register, administer the questionnaire and examine all those aged ≥50 years. We used the standard RAAB questions to collect demographic data, cataract surgical history and data on barriers to cataract surgery. Attempts were made to revisit households of

missing eligible participants before leaving the cluster area. Each survey team completed the assessment of all households in one cluster each day.

## Ophthalmic examination

Examinations were conducted following the RAAB protocol to determine vision status and cause(s) of vision impairment. Presenting visual acuity for each eye was determined outdoors using hand-held Snellen tumbling E charts of 6/12, 6/18 and 6/60 optotypes at 6 meters, and at 3 meters if the 6/60 optotypes could not be seen at 6 meters. Presenting vision impairment status was assigned for each eye and for each person per WHO International Classification of Disease (ICD) 11 criteria as follows: Blindness <3/60; Severe Vision Impairment (SVI) <6/60 to ≥3/60; Moderate Vision Impairment (MVI) <6/18 to ≥6/60; Mild Vision Impairment <6/12 to ≥6/18. Mild Vision Impairment is also referred to as Early Vision Impairment [20] and we used the term Early Vision Impairment (EVI) in this report to prevent confusion between identical abbreviations for Moderate Vision Impairment (MVI) and Mild Vision Impairment (MVI). Best-corrected visual acuity was determined using a pinhole for any eye with presenting acuity <6/12 [22].

Each eye with a presenting acuity <6/12 was examined to assign the cause of vision impairment. If pinhole acuity improved the eye to 6/12, refractive error was determined to be the cause. The crystalline lens was assessed by torch and was graded as normal, obvious lens opacity (cataract), lens absent with or without intraocular lens (IOL) or could not be viewed. Pupils were dilated with 1% tropicamide when vision was <6/12 and the cause of vision loss was not refractive error, cornea or lens-related, so that the posterior pole could be viewed with the direct ophthalmoscope. Causes of impairment for each eye and for each person were determined by the examiner following the WHO convention of assigning the major cause to the primary disorder. When more than one cause was present, the one that was most advanced and easiest to treat was documented as the primary cause [23].

## Training and quality assurance

The survey teams received five days of RAAB training using a standard curriculum by a certified trainer (YS). This included information on how to conduct the survey, collect data, perform quality assurance and carry out pilot testing. Each team consisted of a medical doctor and an experienced ophthalmic technician. Neither doctor was an ophthalmologist, but both doctors had received some ophthalmology training and had several months of work experience in diagnosing and treating eye problems in ophthalmology outpatient clinics. The inter-observer variation for measurement of visual acuity and evaluation of lens status was good for both teams when compared to a senior ophthalmologist (LH) serving as the gold standard (kappa = 0.67 and 0.71 respectively).

## Data management and analysis

All survey data were entered directly into tablet computers using the mobile RAAB (mRAAB) Android application (available at play.google.com), which provides immediate automated range, logic and missing data checks. Data were transferred daily into a study database where additional checks for errors occurred. Standardized data analysis was generated automatically by the RAAB analysis software at the conclusion of the survey. The most recent refugee population data for 10-year age strata and gender groups was obtained from the UN and used in age and sex adjustment.

The automated RAAB software analysis calculates frequency of participation; stratification by sex and 10-year age groups within the sample and within the studied population;

unadjusted and age- and sex-adjusted prevalences of BVI; causes of BVI; cataract surgical coverage (CSC, defined as the proportion who have had cataract surgery to the total of those who have had and those who still need surgery); effective cataract surgery coverage (eCSC, defined as the proportion of people after surgery whose vision is 6/18 or better compared with the total number of people who have had surgery) [24]; WHO-defined vision outcomes after cataract surgery (good ≥6/18, borderline <6/18 to 6/60 and poor <6/60); frequency of reported barriers to cataract surgery; coverage for near and distance refractive correction; and extrapolation of needs to the entire population. Confidence intervals around point estimates were adjusted for the cluster-sampling methodology. Additionally, chi-squared testing was conducted to determine difference between sampled and overall populations, and differences between sexes in categorical variables using OpenEpi software version 3.01 (http://www.openepi.com) with $p < 0.05$ considered as significant.

### Ethical approval

This survey was conducted according to the tenets of the Declaration of Helsinki. Ethical review for the study was provided by the Institutional Review Board of the National Heart Foundation Hospital and Research Institute in Dhaka, Bangladesh (Ref: N.H.F.H.& R. I. 4-14/7/Ad/1012). Due to low literacy rates in the target population, verbal consent was obtained from all study participants after providing an explanation of the study and offering the opportunity to decline participation. No personal identifying information was included in the study database. All participants who required additional eye care were treated at the field site or referred for free services provided in camp clinics or at CBBSH. The dataset for this study is archived in the RAAB repository (http://raabdata.info/).

## Results

Among the 3,800 persons in the sample, 3,629 (95.5%) were examined. One hundred sixty-three (4.28%) were not available for examination and one person (0.0276%) refused to participate. The age and sex distributions of the sampled population differed significantly (P<0.001 for both) from those of the full camp population (Table 1). For this reason, prevalences of BVI are age and sex-adjusted.

### Prevalence and causes of blindness and vision impairment

The unadjusted prevalences of BVI did not differ by sex (Table 2). The age and sex adjusted prevalence of blindness was 2.14% (95% CI:1.7–2.6), SVI was 2.35% (95% CI:1.9–2.8), MVI

**Table 1. Composition of Rohingya refugee camp residents and sampled population age ≥ 50 years.**

| Age Group | Men | | Women | | Total | |
|---|---|---|---|---|---|---|
| | Camp Residents n (%) | Study Sample n (%) | Camp Residents n (%) | Study Sample n (%) | Camp Residents n (%) | Study Sample n (%) |
| 50–59 years | 21,115 (52.4) | 1,069 (62.2) | 21,796 (59.4) | 1,436 (75.1) | 42,911 (55.8) | 2,505 (69.0) |
| 60–69 years | 12,905 | 436 | 10,356 | 342 | 23,261 | 778 |
| | (32.0) | (25.4) | (28.2) | (17.9) | (30.2) | (21.4) |
| 70–79 years | 4,977 | 169 | 3,246 | 102 | 8,223 | 271 |
| | (12.4) | (9.83) | (8.85) | (5.33) | (10.7) | (7.46) |
| 80+ | 1,295 | 44 | 1,272 | 31 | 2,567 | 75 |
| years | (3.21) | (2.56) | (3.47) | (1.62) | (3.33) | (2.07) |
| Totals | 40,292 | 1,718 | 36,670 | 1,911 | 76,962 | 3,629 |
| (% by gender) | (52.4) | (47.3) | (47.6) | (52.7) | (100) | (100) |

**Table 2. Sampled prevalence and age and sex adjusted prevalence of presenting vision acuity in the better eye age ≥ 50 years.**

| | | Observed Prevalence in Sampled Population | | | Age and Sex Adjusted Prevalence | |
|---|---|---|---|---|---|---|
| | Men (n = 1,718) | Women (n = 1,911) | Total (n = 3,629) | Men | Women | Total |
| Vision AcuityCategory | n % (95% CI) | n % (95% CI) | n % (95% CI) | % (95% CI) | % (95% CI) | % (95% CI) |
| Blindness | 27 | 33 | 60 | | | |
| | 1.57 | 1.73 | 1.65 | 1.91 | 2.40 | 2.14 |
| | (0.9–2.2) | (1.1–2.3) | (1.2–2.1) | (1.2–2.6) | (1.8–3.0) | (1.7–2.6) |
| SVI | 28 | 34 | 62 | | | |
| | 1.63 | 1.78 | 1.71 | 1.86 | 2.89* | 2.35 |
| | (1.0–2.3) | (1.2–2.3) | (1.3–2.1) | (1.2–2.5) | (2.3–3.4) | (1.9–2.8) |
| MVI | 130 | 152 | 282 | | | |
| | 7.57 | 7.95 | 7.78 | 8.68 | 10.8** | 9.68 |
| | (6.0–10.0) | (6.2–9.8) | (6.4–9.2) | (7.1–10.3) | (9.0–12.6) | (8.3–11.1) |
| EVI | 246 | 249 | 495 | | | |
| | 14.3 | 13.0 | 13.6 | 15.1 | 14.2 | 14.7 |
| | (11.5–17.1) | (10.4–15.6) | (11.2–16.0) | (12.3–17.9) | (11.6–16.8) | (12.3–17.1) |

CI = confidence interval.

SVI = severe vision impairment.

MVI = moderate vision impairment.

EVI = early vision impairment.

* indicates prevalence differed between men and women p = 0.023.

** indicates prevalence differed between men and women p = 0.018.

was 9.68% (95% CI: 8.3%-11.1) and EVI was 14.7% (95% CI: 12.3–17.1). The adjusted prevalence of blindness and adjusted prevalence of EVI did not differ by gender. Women were more likely to have SVI compared to men (2.89% vs. 1.86%, p = 0.023) and were more likely to have MVI compared to men (10.8% vs. 8.68%, p = 0.018).

Cataract was the primary cause of blindness (45/60 = 75.0%) and of SVI (47/62 = 75.8%), while refractive error was the leading cause of MVI (136/282 = 48.2%) and of EVI (453/495 = 91.5%). Avoidable causes were responsible for 78.3% (n = 47/60) of all blindness and 95.9% (n = 862/899) of all BVI (Table 3).

## Cataract Surgical Coverage (CSC) and vision outcomes

The rate of CSC was 81.82% among blind persons, and 72.45% among blind eyes (Table 4). Men had higher CSC rates by person and by eye compared to women at all acuity levels, but these differences were only significant by person at <6/18 (p = 0.013) and by eye at <6/60 (p = 0.040) and at <6/18 (p <0.001). The eCSC rates by person for men were likewise all higher at all acuity levels than for women but these differences were only significant at <3/60 (p = 0.032) and at <6/18 (p = 0.001).

Nearly all cataract surgeries (99.6%, n = 262/263) were performed with the use of intraocular lenses (IOLs), and a majority of surgeries (83.7%, n = 220/263) had been done at CBBSH. Most operated eyes (181/263 = 68.8%) had good presenting vision outcomes (≥6/18). Among the 32 eyes (12.2%) with poor outcomes, 25 (78.1%) were due to chronic complications. Among all 263 operated eyes, 70 eyes (26.6%) could have improved vision with appropriate spectacles and 61 eyes (23.2%) had long-term complications that reduced vision. The

**Table 3. Causes of presenting blindness and vision impairment in best-seeing eye of sampled population.**

| Cause of Vision Loss | Blind < 3/60 | Severe Vision Impairment ≤ 6/60 > 3/60 | Moderate Vision Impairment ≤ 6/18 > 6/60 | Early Vision Impairment ≤ 6/12 > 6/18 |
|---|---|---|---|---|
| | n (%) | n (%) | n (%) | n (%) |
| Cataract untreated | 45 (75.0) | 47 (75.8) | 119 (42.2) | 30 (6.06) |
| Other Posterior | 8 (13.3) | 8 (12.9) | 9 (3.19) | 2 (0.404) |
| All other globe / Central Nervous System | 4 (6.66) | 0 | 1 (0.354) | 0 |
| Cornea opacity | 1 (1.66) | 1 (1.61) | 2 (0.709) | 3 (0.606) |
| Age Related Macular Degeneration | 1 (1.66) | 1 (1.61) | 2 (0.709) | 1 (0.202) |
| Other (Glaucoma, Phthisis) | 1 (1.66) | 0 | 1 (0.354) | 1(0.202) |
| Refractive error | 0 | 3 (4.83) | 136 (48.2) | 453 (91.5) |
| Myopic Degeneration | 0 | 2 (3.22) | 6 (2.12) | 2 (0.404) |
| Cataract Surgery Complications | 0 | 0 | 6 (2.12) | 3 (0.606) |
| Totals | 60 (100) | 62 (100) | 282 (100) | 495 (100) |
| All treatable causes | 45 (75.0) | 50 (80.7) | 255 (90.4) | 483 (97.6) |
| All preventable causes | 2 (3.33) | 3 (4.83) | 15 (5.32) | 9 (1.82) |
| All avoidable causes | 47 (78.3) | 53 (85.5) | 270 (95.7) | 492 (99.4) |

Treatable causes = cataract & refractive error.

Preventable causes = cornea opacity, phthisis, cataract surgery complications, glaucoma.

Avoidable causes = treatable + preventable causes.

proportion of eyes with good outcomes did not differ significantly between those operated in the last 3 years (65.2%, n = 105/161) and those operated longer than 7 years ago (75.0%, n = 33/44) [p = 0.112].

## Barriers to uptake of cataract surgery

A total of 95 participants (2.62%) were blind or had SVI due to untreated bilateral cataract. Their reasons for not seeking surgery included 'need not felt' (33.7%, n = 32), 'cost' (30.5%, n = 29), 'fear' (16.8%, n = 16), 'treatment denied by provider' (10.5%, n = 10), 'cannot access treatment' (5.26%, n = 5), and 'unaware treatment was possible' (3.15%, n = 3).

**Table 4. Cataract Surgical Coverage (CSC) and effective Cataract Surgical Coverage (eCSC) in sampled population by person and by eye.**

| Best Corrected Vision Acuity (BCVI) determined by pinhole | Men % | Women % | Total % | p-value |
|---|---|---|---|---|
| BCVI < 3/60 | | | | |
| CSC by person | 85.29 | 78.13 | 81.82 | 0.149 |
| eCSC by person | 69.11 | 53.12 | 61.36 | 0.032 |
| CSC by eye | 75.13 | 69.28 | 72.45 | 0.109 |
| BCVI < 6/60 | | | | |
| CSC by person | 69.77 | 66.67 | 68.29 | 0.337 |
| eCSC by person | 56.98 | 44.87 | 51.22 | 0.063 |
| CSC by eye | 64.35 | 55.09 | 60.46 | 0.040 |
| BCVI < 6/18 | | | | |
| CSC by person | 55.08 | 41.04 | 47.62 | 0.013 |
| eCSC by person | 45.76 | 27.61 | 36.11 | 0.001 |
| CSC by eye | 48.52 | 35.94 | 42.08 | <0.001 |

### Refractive error and presbyopia coverage

Refractive error was detected in 17.1% (n = 622/3,629) of participants and 95.2% (n = 592/622) of these did not have spectacles. The spectacle coverage rate for men (5.28% n = 16/303) and women (4.39% n = 14/319) were not significantly different (p = 0.31). All participants were ≥50 years of age and would be expected to require presbyopic correction for reading or other near work, and women were significantly more likely to have a presbyopic correction compared to men, 5.44% (n = 104/1,911) vs. 3.96% (n = 68/1,718) [p = 0.018].

### Extrapolation to population

Extrapolating the findings in this study to the entire Rohingya displaced population, we estimate that 26.7% (n = 20,561) of the 76,962 Rohingya refugees aged ≥50 years have vision impairment (SVI+MVI+EVI) and an additional 2.14% (n = 1,645) are blind (Table 2). Correcting all eyes that are blind or have vision acuity <6/60 or <6/18 due to cataract, an estimated 2,718, 4,753 and 9,962 surgeries respectively would be required. Furthermore, over 12,000 Rohingya refugees would require correction of distance refractive error and an additional 73,000 would require reading glasses.

## Discussion

To the best of our knowledge, this is the first reported population-based data on prevalence of BVI in the adult Rohingya population. To the best of our knowledge, this is also the first RAAB conducted in a UNHCR refugee camp setting. We found a 2.14% age and sex-adjusted prevalence of blindness among those aged 50 years and above. Seventy-five percent of blindness and 95.8% of all BVI was avoidable, most of which were due to cataract or uncorrected refractive error. The prevalence of blindness among the Rohingya refugees was expected to be higher but was in fact comparable to that reported in previous studies among nearby populations in both Bangladesh and Myanmar. A series of RAABs conducted in 8 districts in Bangladesh, including Cox's Bazar, had reported blindness rates ranging from 0.5%–4.0% with a mean prevalence of 2.2% [21]. Unpublished RAABs undertaken in two rural areas of Myanmar reported blindness of 2.3% and 2.9%.

Population data on BVI from refugees and other conflict-affected populations are limited, varying and not always comparable because of differences in methodology. For example, the prevalence of blindness was 3.4% in the 2010 RAAB survey carried out in occupied Palestine Territories [6] and 2.1% in a non-RAAB all-age survey of Afghan refugees in Pakistan in 1998 [5], though the latter figure is not comparable to the present study due to age differences in the sampled population. The prevalence of blindness was reported at 1.1% and 1.8% in RAAB studies in post-conflict populations in Burundi and Rwanda in 2012 and 2007 respectively [7, 8]. The population prevalence of blindness was found to be 4.1% in a non-RAAB study in 2006 among persons aged 5 and above in conflict-affected South Sudan [9]. The prevalence of blindness in South Sudan was very high in part due to lack of eye services combined with endemic blinding trachoma [10] while the low prevalences reported in Rwanda and Burundi are partially explained by the comparatively low proportion of the population aged 60 years and above, among whom blindness is more common.

We also noted a comparatively lower proportion of elderly persons in our study cohort, with only 3.6% of the Rohingya refugee population being aged 60 years and above [11], compared to 7.2% in Bangladesh and 8.5% in Myanmar [25]. The lack of older persons among the Rohingya refugees in the camps may be due to a lower life expectancy associated with difficult living conditions, unwillingness of the elderly to flee Myanmar, or low rates of survival during the journey. Given that mortality rates are higher in people blind or visually impaired [26–28]

elderly Rohingya with vision impairment would be even less likely to survive than their non-impaired peers. These factors likely contribute to rates of BVI being lower than expected among the Rohingya refugees.

Considering this population lacked access to eye care prior to coming to the refugee camps, cataract surgery coverage for those with blinding cataracts was high (81.82%, Table 4). Most of these surgeries 83.7% occurred at CBBSH, the only eye hospital in Cox's Bazar and the presenting vision outcomes were generally good (68.8% ≥6/18 in the operated eye). Similar good surgical results, as well as high uptake of eye services by the displaced Rohingya, was reported in a recent clinic-based study in this population [15]. The good surgery coverage provided by CBBSH combined with reasonably good vision outcomes also likely contribute to rates of BVI being lower than expected among the Rohingya refugees.

Strengths of this study include using the well-established RAAB protocols [19], and a high participation rate. The prevalence of BVI were age and sex adjusted to correct for differences between the sampled and camp populations. Standardization of data management and analysis helps ensure that RAAB results are comparable across studies. This study is not without limitations. The RAAB methodology does not include participants <50 years old because a majority of blindness is found in older adults [19], thus determining rates and causes of BVI among those under age 50 would require additional surveying beyond this RAAB. However, clinical experience with the Rohingya suggests that blinding and visually significant cataract is a burden among adults of all ages including those under age 50 [15]. Further, the RAAB approach does not provide a detailed diagnosis of posterior segment conditions [19], which may lead to misclassification of such diseases. Our use of non-ophthalmologists as examiners may have further limited ability to correctly diagnose posterior segment conditions. An optional diabetic retinopathy (DR) module for the RAAB survey is available [20] and we could further improve our knowledge of this unique population, particularly with regards to diabetic eye diseases, by planning a RAAB+DR in the future.

These data will guide eye health system strengthening efforts in Cox's Bazar and provide a baseline against which to measure progress in eye care delivery and outcomes among the Rohingya refugees. Our study reveals several areas where improvements are needed in eye service delivery for the Rohingya. Cataract surgery coverage has room for improvement and vision outcomes, while reasonably good, have not yet reached WHO suggested standards. Refractive error and presbyopia were rarely corrected, and services will need to scale up significantly to deliver suitable spectacle coverage. Compared to men, the Rohingya women had higher rates of severe and moderate vision impairment (SVI & MVI, Table 2) and lower rates of effective cataract surgery coverage (eCSC) at <3/60 and at < 6/18 (Table 4). Efforts to address gender disparity are being included in eye program planning. Additionally, research will be needed to determine rates and causes of BVI in Rohingya refugee population that is under 50 yrs. of age.

In conclusion, this RAAB survey in the Rohingya refugees currently settled in Bangladesh showed that the need for eye care in these communities justifies the inclusion of eye services in the provision of Rohingya health care. We suggest the United Nations High Commissioner for Refugees (UNHCR) and the NGOs providing health care to refugees include eye care such as cataract surgery and refractive error and presbyopia correction in the health service for all refugees with long-term displacement.

## Supporting information

**S1 File. Strengthening the Reporting of Observational Studies in Epidemiology (STROBE) checklist.**
(PDF)

## Acknowledgments

We would like to thank the office of the Refugee Relief and Repatriation Commissioner (RRRC), and the Mahji refugee leaders for their patience, understanding, and support in undertaking this survey. Thanks to Hans Limburg for valuable input on data management.

This survey was commissioned by the Bangladesh Ministry of Health and Family Welfare (MOHFW) and was conducted by Orbis International, CSF Global, the Seva Foundation and the International Agency for the Prevention of Blindness (IAPB). The donor of the study had no role in study design, data collection, data analysis, data interpretation, or writing of the report.

An abstracted presentation of these findings was presented at the annual Council of Members meeting of the International Agency for the Prevention of Blindness at Dar Es Salaam, Tanzania in October 2019.

## Author Contributions

**Conceptualization:** A. H. M. Enayet Hussain, Munir Ahmed, Jerry E. Vincent, Yuddha D. Sapkota, Taraprasad Das, Lutful Husain, Danny Haddad.

**Data curation:** Jerry E. Vincent, Johurul Islam, Yuddha D. Sapkota, Lutful Husain, Mohammad Awlad Hossain.

**Formal analysis:** Jerry E. Vincent, Yuddha D. Sapkota, Lutful Husain.

**Funding acquisition:** A. H. M. Enayet Hussain, Munir Ahmed.

**Investigation:** A. H. M. Enayet Hussain, Munir Ahmed, Jerry E. Vincent, Johurul Islam, Yuddha D. Sapkota, Lutful Husain, Mohammad Awlad Hossain.

**Methodology:** A. H. M. Enayet Hussain, Munir Ahmed, Jerry E. Vincent, Johurul Islam, Yuddha D. Sapkota, Nathan Congdon, Lutful Husain, Gulam Khandaker, Mohammad Muhit, Danny Haddad.

**Project administration:** A. H. M. Enayet Hussain, Munir Ahmed, Johurul Islam, Gulam Khandaker, Mohammad Muhit.

**Supervision:** Munir Ahmed, Johurul Islam, Lutful Husain, Mohammad Awlad Hossain.

**Validation:** Yuddha D. Sapkota, Lutful Husain.

**Visualization:** Jerry E. Vincent, Yuddha D. Sapkota, Taraprasad Das, Nathan Congdon.

**Writing – original draft:** A. H. M. Enayet Hussain, Munir Ahmed, Jerry E. Vincent, Johurul Islam, Yuddha D. Sapkota, Taraprasad Das, Danny Haddad.

**Writing – review & editing:** A. H. M. Enayet Hussain, Munir Ahmed, Jerry E. Vincent, Johurul Islam, Yuddha D. Sapkota, Taraprasad Das, Nathan Congdon, Lutful Husain, Gulam Khandaker, Mohammad Muhit, Mohammad Awlad Hossain, Danny Haddad.

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
