## [Decision Letter · Decision Letter 0]

29 Oct 2020

PONE-D-20-26584

Rapid Assessment of Avoidable Blindness and Cataract Surgery Coverage among Forcibly Displaced Myanmar Nationals (Rohingya Refugees) in Cox’s Bazar, Bangladesh

PLOS ONE

Dear Dr. Jerry E Vincent,

Thank you for submitting your manuscript to PLOS ONE. After careful consideration, we feel that it has merit but does not fully meet PLOS ONE’s publication criteria as it currently stands. Therefore, we invite you to submit a revised version of the manuscript that addresses the points raised during the review process.

We look forward to receiving your revised manuscript.

Kind regards,

Fakir M Amirul Islam, PhD

Academic Editor

PLOS ONE

Additional Editor Comments (if provided):

This is a very well written manuscript. Importantly, the study was needed to save sight among one of the most disadvantaged population. The study could lead to an impactful outcome.

Some minor comments:

There are some grammatical errors. Formatting of the Tables could be better. Please pay attention to formatting a bit. Another example,

Camp Residents

N (%)

21,115

(52.4%), just 52.4, please remove % throughout in Tables 1-4. Please do better formatting.

Since the % is given on the title, it does not need to repeat in the Tables. Please remove the percentage from all the Tables while this is given on the title.

Women were more likely to have SVI (2.89% vs.1.86%, p = 0.0229) and MVI compared to men (10.8% vs. 8.68%, p =0.0184) compared to men.

** The P value could be 2-3 decimals.

Reviewers' comments:

Reviewer's Responses to Questions

**Comments to the Author**

1. Is the manuscript technically sound, and do the data support the conclusions?

Reviewer #1: Yes

Reviewer #2: Yes

2. Has the statistical analysis been performed appropriately and rigorously? 

Reviewer #1: Yes

Reviewer #2: Yes

3. Have the authors made all data underlying the findings in their manuscript fully available?

Reviewer #1: Yes

Reviewer #2: Yes

4. Is the manuscript presented in an intelligible fashion and written in standard English?

Reviewer #1: Yes

Reviewer #2: Yes

5. Review Comments to the Author

Reviewer #1: This paper presents very interesting and novel findings. I'm not aware of any studies reporting similar data in a disadvantaged population.

These data are very important to some extents:

1. Showing that RAAB methods are universally applicable to many settings

2. Similar study should be performed in other disadvantaged populations.

I have no further comments. The methods are sound, well-written and the flow of the paper is smooth.

Some minor grammatical errors are noted, and perhaps demand further revision.

Reviewer #2: 1. This is a useful study in which the authors make comparisons to other displaced / refugee populations.

Given the younger age group, can the authors comment on the prevalence of other diseases such as diabetic retinopathy that may occur in younger populations.

2. What improvements could be made to a future study given the observations found in this study. For example, Should RAAB-DR model be considered?

6. PLOS authors have the option to publish the peer review history of their article (what does this mean?). If published, this will include your full peer review and any attached files.

Reviewer #1: No

Reviewer #2: **Yes: **Rahul Chakrabarti

---

## [Author Response · Author response to Decision Letter 0]

11 Nov 2020

Multiple authors have reviewed the manuscript for grammatical errors and additionally, the manuscript was reviewed with “Grammarly” a grammar checking application (available at www.grammarly.com). We have found no errors.

Recognizing that grammar use, and grammar rules vary, if the editor or reviewers feel that specific grammar errors are present in our manuscript, we would be happy to address them as needed.

We have gone through Tables 1-4 and have removed the percentage sign (%) as requested from the body of the tables. Tables 1- 4 are on page 9, 10, 11 and 13 respectively. 

We reviewed all p values and edited as needed. No p values are now longer than 3 decimal places. Edits were made for p values in Table 2 (footnotes) and Table 4 (pages 10 and 13 respectively). Edits for p values were made in the text at lines 211, 212, 543, 544 and 546 in the Track Change version which correspond to lines 211, 212, 247, 248 and 250 in the corrected final manuscript. 

Reviewer One does not provide specific examples or lines for grammatical errors. As noted earlier, multiple authors have reviewed the manuscript for grammatical errors and additionally, the manuscript was reviewed with “Grammarly” a grammar checking application (available at grammarly.com). We have found no errors.

Recognizing that grammar use, and grammar rules vary, if the editor or reviewers feel that specific grammar errors are present in our manuscript, we would be happy to address them as needed.

On the basis of these two comments we have added to the study limitations discussion. That the RAAB methodology does not collect prevalence data on those under age 50 is a standard limitation and we added emphasis to this in the study limitations discussion. We also noted availability of DR module for the RAAB and that this would address some of the limitations on posterior segment disease. Our original text is in italic and the additional text is in ALL CAPS. 

“This study is not without limitations. The RAAB methodology does not include participants <50 years old because a majority of blindness is found in older adults [19], THUS DETERMINING RATES AND CAUSES OF BVI AMONG THOSE UNDER AGE 50 WOULD REQUIRE ADDITIONAL SURVEYING BEYOND THIS RAAB. However, clinical experience with the Rohingya suggests that blinding and visually significant cataract is a burden among adults of all ages including those under age 50 [15]. Further, the RAAB approach does not provide a detailed diagnosis of posterior segment conditions [19], which may lead to misclassification of such diseases. Our use of non-ophthalmologists as examiners may have further limited ability to correctly diagnose posterior segment conditions. AN OPTIONAL DIABETIC RETINOPATHY (DR) MODULE FOR THE RAAB SURVEY IS AVAILABLE [20] AND WE COULD FURTHER IMPROVE OUR KNOWLEDGE OF THIS UNIQUE POPULATION, PARTICULARLY WITH REGARDS TO DIABETIC EYE DISEASES, BY PLANNING A RAAB+DR IN THE FUTURE.”

These changes are found on lines 726-727 and 732-735 in the Track Changes version, corresponding to lines 362-363 and 368-371 in the corrected final manuscript.

In addition to the above responses, we have also located and corrected the following typographical errors:

• Name spelling correction on line 51 from Baituch to Baitush

• Symbol correction on line 172 (manuscript) / line 173 (track changes) from ≤6/60 to <6/60 

• Name spelling on lines 431- 432 (manuscript) / lines 796 - 797 (track changes) from Lutful Hussain to Lutful Husain.

---

## [Editor Report · Decision Letter 1]

13 Nov 2020

Rapid Assessment of Avoidable Blindness and Cataract Surgery Coverage among Forcibly Displaced Myanmar Nationals (Rohingya Refugees) in Cox’s Bazar, Bangladesh

PONE-D-20-26584R1

Dear Dr. Vincent,

We’re pleased to inform you that your manuscript has been judged scientifically suitable for publication and will be formally accepted for publication once it meets all outstanding technical requirements.

Kind regards,

Fakir M Amirul Islam, PhD

Academic Editor

PLOS ONE

Additional Editor Comments (optional):

The  minor comments are addressed. Congratulations for this !!
---

## [Editor Report · Acceptance letter]

19 Nov 2020

PONE-D-20-26584R1 

Rapid Assessment of Avoidable Blindness and Cataract Surgery Coverage among Forcibly Displaced Myanmar Nationals (Rohingya Refugees) in Cox’s Bazar, Bangladesh 

Dear Dr. Vincent:

I'm pleased to inform you that your manuscript has been deemed suitable for publication in PLOS ONE. Congratulations! Your manuscript is now with our production department. 

Kind regards, 

on behalf of

Dr Fakir M Amirul Islam 

Academic Editor

PLOS ONE